# CODA: A Design Assistant to Facilitate Specifying Constraints and Parametric Behavior in CAD Models

**Tom Veuskens, Florian Heller, Raf Ramakers**

Hasselt University - tUL - Flanders Make, Expertise Centre for Digital Media, Diepenbeek, Belgium

firstname.lastname@uhasselt.be

## ABSTRACT

We present CODA, an interactive software tool, implemented on top of Autodesk Fusion 360, that helps novice modelers with designing well-constrained parametric 2D and 3D CAD models. We do this by extracting relations that are present in the design but are not yet enforced by constraints. CODA presents these relations as suggestions for constraints and clearly communicates the implications of every constraint by rendering animated visualizations in the CAD model. CODA also suggests dynamic alternatives for static constraints present in the model. These suggestions assist novices in CAD to work towards well-constrained models that are easy to adapt. Such well-constrained models are convenient to modify and simplify the process to make design alternatives to accommodate changing needs or specific requirements of machinery for fabricating the design.

**Keywords:** CAD modeling, constraints, novices, end-user modeling, fabrication.

**Index Terms:** Human-centered computing—Interactive systems and tools Human-centered computing—Visualization—Visualization techniques;

## 1 INTRODUCTION

The maker movement is largely driven by a community of DIY enthusiasts building on each other's work by sharing digital versions of artefacts through online platforms, such as Thingiverse[1] or Youmagine[2] [8, 34]. Makers frequently do this by making rough edits to mesh files or by starting from scratch while using concepts from existing designs [9]. A more convenient way to adapt an existing model is to adjust parameters in a well-constrained parametric model. Well-constrained parametric models allow for dimensional adjustments and personal and aesthetic refinements. Additionally, such changes can also make models compatible with machines and materials available in other labs, for example by compensating for shrinkage of ABS 3D printing filament or the kerf of a laser cutter. González-Lluch and Plumed [10] show however, that even for engineering students it is challenging to verify whether 3D models are fully constrained and thus behave as desired when changes are made. Especially under time pressure, trained CAD modelers produce 3D models that are hard to modify because of errors or missing constraints [36]. For makers this can become even more challenging as many do not have a formal training in CAD modeling [23].

To empower and encourage makers to design and share parametric models, Thingiverse launched a Customizer platform [30] that allows for making adjustments to parametric 3D models using simple GUI controls, such as sliders and drop-down menus. However, Thingiverse Customizer requires making the entire 3D model in a CSG scripting language [35], which is significantly different from the feature-based modeling approach supported by popular 3D CAD environments [15], such as Autodesk Fusion and SolidWorks. An analysis of Thingiverse in 2015 shows that only a small fraction of designs (1.0%) are compatible with the Customizer [1].

To help users in specifying parametric models, recent versions of 3D CAD environments automatically add basic constraints to 2D sketches. AutoCAD's auto-constraint feature [20] and BricsCAD's Auto Parameterize functionality [12] go even further and automatically inject constraints in models. As there are multiple valid alternatives to constrain models, fully-automatically introducing constraints does not always lead to the desired behavior of the

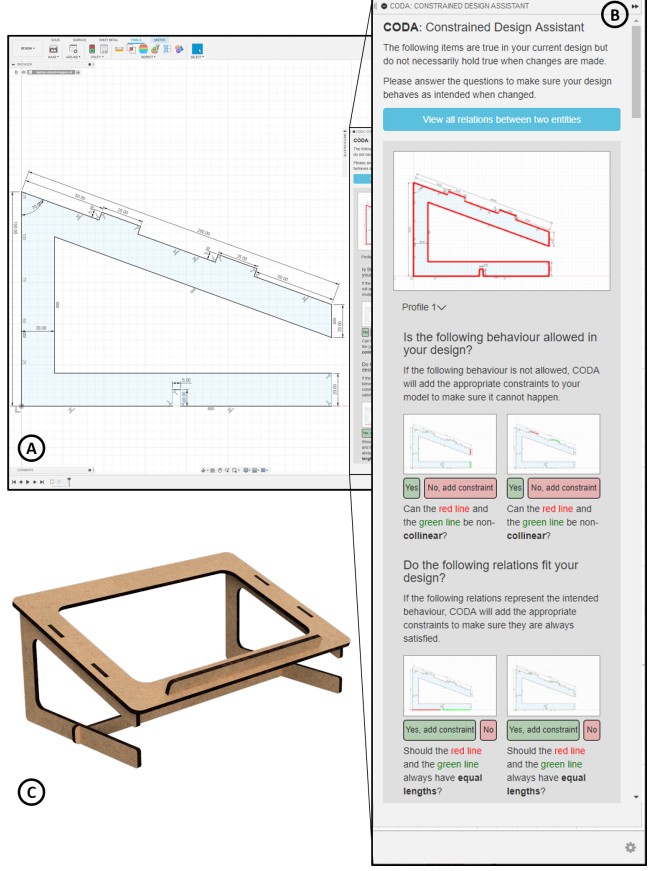

Figure 1: Using CODA to correctly specify parametric behavior in a 3D CAD model of a laptop stand. (a) A 2D sketch of the cross-sectional profile for the laptop stand designed by the user. (b) CODA lists relations that are present in the design but not yet enforced by constraints. (c) Accepting CODA's suggestions allows novice modelers to quickly transition to a well-constrained model ready for fabrication.

---

[1] https://www.thingiverse.com/

[2] https://www.youmagine.com/

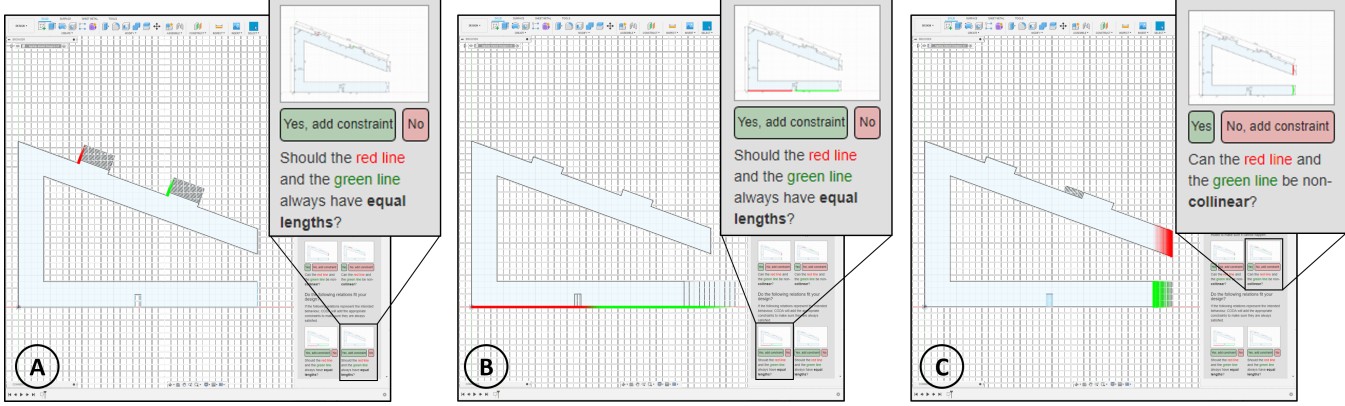

Figure 2: CODA's animations show the implications of suggested constraints. (a) A suggestion for constraining the two tabs to have the same height. (b) A suggestion for overwriting a constraint that ensures the slot at the bottom is always centered. (c) A suggestion for making the front edges collinear. (All three figures are edited to visualize the animation)

model. For example, the hole for the charging cable of a phone holder (Figure 3) is positioned in the center of the design. It is up to the designers preferences whether to constrain this hole at a fixed distance or a ratio from the top or the bottom of the phone holder. Therefore, the system presented in this paper takes a different approach and suggests constraints and explains their differences and implications on the model.

In this paper, we present CODA, a **Co**nstrained **D**esign **A**ssistant. CODA is an interactive software tool that helps novice CAD modelers with designing well-constrained 2D and 3D CAD models. Such well-constrained models are convenient to modify which simplifies the process to make design alternatives or when making adjustments for fabricating models with different machinery. First, our system makes users aware of relations that are present in the current design but are not yet enforced by constraints, such as edges being parallel without a parallel constraint. Second, CODA reconsiders static constraints added by the user and suggests more dynamic alternatives to make the design flexible to changes. CODA also helps in communicating the meaning and implications of all suggested constraints by animating the model and demonstrating its implications (Figure 2).

The core contribution of this paper is *CODA*, an interactive software assistant to aid novice modelers in making well-constrained CAD models that are robust to changes. More specifically, we contribute:

1. A computational approach for extracting relations in a model which are not yet enforced by constraints.
2. A set of novel interactive animations to clearly communicate the impact of constraints to novice users.

## 2 RELATED WORK

This work draws from, and builds upon prior work on facilitating CAD modeling and work related to sharing models for fabrication.

### 2.1 Facilitating CAD Modeling

3D CAD modeling environments offer hundreds of features. How these features are used and combined determine the flexibility, adaptability, and ultimately the reusability of 3D models [5, 11]. Research shows, however, that even models designed by students with a formal 3D modeling training are often hard to reuse and adapt, especially when designed under time pressure [36]. In line with these observations, González-Lluch and Plumed [10] show that engineering students have a hard time reasoning whether profiles are over- or under-constrained. When considering modeling within the maker

community, an emerging group of people learn 3D modeling practices by themselves through online resources [23]. However, these novice modelers could significantly benefit from high-quality models that are easy to adapt as they frequently make new artefacts by starting from existing designs [9].

To lower the barrier to get started with 2D and 3D modeling, various tools have been developed that specifically target novices, such as Autodesk Tinkercad[3] and BlocksCAD[4]. However, several studies with casual makers [17, 31], children [18], and students in special education schools [4] show that 3D modeling is still challenging. To facilitate further, the Chateau [19] system helps with CAD modeling by suggesting modeling operations based on simple sketch gestures by the user. Rod et al. [38] presents various novel interaction techniques to further facilitate 3D modeling on touch-screen devices.

Instead of adapting CAD environments and making custom modeling operations for novices, researchers also explored how to facilitate the process for novices to learn a new CAD environment. GamiCAD [28] and CADament [29] gradually introduce sketching and modeling operations using gamification techniques to lower the barrier and keep novices motivated to continue learning new aspects. Alternatively, Blocks-to-CAD [25] shows how to gradually introduce 3D modeling operations in sandbox games, such as Minecraft, to get newcomers introduced to the basics of CAD modeling. Additionally, recent research results show how modeling strategies from experts can be modeled, analyzed, and compared to provide guidance for other users during modeling sessions [6].

Instead of embedding expert knowledge in software systems, software systems can help in bringing novices in contact with experts while facing issues with 3D modeling. MicroMentor [21], for example, makes it possible for novices to request one-on-one help for specific issues. In contrast, the Maestro [7] system makes educators in workshops aware of student's progress and common challenges as they occur. Although experts typically provide more nuanced answers to the various challenges novices face, experts often need an incentive to help other users and first need to get familiar with the specific problem the user is facing [21].

Several techniques have been developed that specifically aim to improve the adaptability and re-usability of models by facilitating specifying parametric behaviour. PARTs [16] allows users to embed additional geometry into the design to specify how a model is

---

[3]https://www.tinkercad.com/
[4]https://www.blockscad3d.com/

supposed to be used. The dimensions of the additional geometry dictate the dimensions of the PART (e.g. by specifying x amount of material needs to surround the geometry). In contrast, CODA takes a more generic approach that is also applicable on models that are not driven by real world objects and allows for general-purpose parametric specifications. On the other hand, commercial feature-based CAD modeling tools support, for example, snapping interaction techniques to ease and improve precision in 2D sketches, such as centering a point in the middle of a line or sketching two perpendicular lines. Leveraging this snapping functionality often-times automatically fixes the relation by injecting the associated constraints in the 2D sketch. AutoConstraint [20] takes a different approach and adds constraints to a completed sketch until it is fully constrained. Closest to our work is the Auto Parameterize [12] feature of BricsCAD[5] which automatically converts all static dimensions of a 3D model to algebraic equations to facilitate scaling and adapting the model. However, there are always multiple valid alternatives to constrain models and fully-automatically introducing constraints does not always lead to the desired behavior of the model. We therefore take a different approach and present various geometry constraints and algebraic relations that could be applied to the model. CODA communicates the implications of all suggestions using in-context animations to allow novice users to make informed decisions.

Also related to our work are computational approaches to reverse engineering CAD models from mesh models by extracting modeling features [37, 42]. Several systems also present algorithms to detect and extract geometry constraints in mesh models using numerical methods for constrained fitting [2] and techniques for detecting repeating patterns [26] and symmetries [27]. Willis et al. [43] use machine learning techniques to reverse-engineer CAD operations from a CAD model. While these approaches convert mesh models to CAD files, they do not analyze the model to add parametric features or constraints as offered by CODA.

## 2.2 Sharing Models for Fabrication

Over the past decade, digital fabrication has become accessible mainly via public maker labs and affordable digital fabrication equipment [33]. Shewbridge et al. [40] report that households are interested in replacing, modifying, customizing, repairing, or replicating household objects using digital fabrication machinery. However, starters frequently need help from more experienced users to translate ideas into 3D models. This is often done via drawings, photographs, and spoken language [40]. While platforms, such as Upwork[6] and Cad Crowd[7] are available to outsource 3D modeling work, they require additional expenses.

Instead of designing CAD models from scratch, makers often adapt or combine existing 3D models, found on public repositories, such as Thingiverse [4, 9, 17]. This process can be challenging as many users only share triangular mesh file-formats (STL) [1]. While users can request changes for models through the comments section, studies show that only 32% of such requests are granted [1]. To empower novice CAD modelers to adapt models themselves, Thingiverse introduced the Customizer feature [30], a plugin that exposes GUI controls to adjust the parameters of models designed with the OpenSCAD [35] scripting language. While the Thingiverse Customizer is highly popular [8, 34], only a small portion (3.7%) of 3D models available on the platform are modeled in OpenSCAD, and only 1% are compatible with the Customizer [1]. Hudson et al. [17] observe that modeling in OpenSCAD is challenging and significantly different from feature-based parametric modeling environments traditionally used by CAD modelers.

---

[5]https://www.bricsys.com/
[6]https://www.upwork.com/
[7]https://www.cadcrowd.com/

In contrast to these efforts, CODA guides and stimulates novice modelers in making well-constrained models in popular feature-based CAD modeling environments. Well-constrained parametric models are convenient to adapt as they represent a family of alternative models [3].

## 3 SYSTEM OVERVIEW

This section gives an overview of CODA's core features. We start with a short walkthrough demonstrating how our system can be used in a real modeling workflow. Afterwards, we discuss CODA's features in more detail.

### 3.1 Walkthrough

This walkthrough demonstrates the design process Emily, a novice modeler in Autodesk Fusion 360, follows to design a laptop stand that can be laser cut (Figure 1c). During this process, CODA offers support to make a laptop stand that is well-constrained, and easy to adapt and scale to other laptops or devices (Figure 1b).

As shown in Figure 1a, Emily starts with sketching the 2D cross-sectional profile for the laptop stand. She adds dimensions to the sketch to fit the size of her laptop. While sketching, CODA informs Emily that the slot and tabs are 5mm in size and asks whether these features should always have the same dimension. When hovering this suggestion, CODA animates the model by resizing these features at the same time to demonstrate the effect of the suggested constraint (Figure 2a). Emily accepts the suggested constraint and CODA replaces the static dimension constraint with dimensions that share the same value (variable).

CODA also notices that the slot at the bottom is currently positioned in the center but not constrained as such. Therefore the system suggests to replace the dimension that offsets the slot from the left edge with a constraint that ensures that the bottom edges on both sides of the slot are always equal. Again, Emily accepts the suggestion after inspecting the animation to understand the implications of the constraint (Figure 2b).

Next, Emily notices a suggestion for making the two vertical edges at the right of the profile collinear. When hovering the suggestion, the animation informs her that when the size of the slanted edge of the stand would change, the bottom of the laptop stand does not yet adjust accordingly (Figure 2c). Emily accepts the suggestion to make the two edges collinear as she prefers a laptop stand that is well aligned. CODA offers more relevant suggestions to improve the constraints in this cross-sectional profile which Emily can accept as desired. Examples include, making all three sides equally wide (uniform thickness), relating the width of the two tabs, and suggestions related to the positioning of the two tabs.

Further in the design process, when extruding the profiles 5mm, CODA also notices this extrusion depth equals the size of the tabs and slots and suggests creating a constraint. When the final laptop stand is finished, Emily can easily adjust the stand to fit other laptops or adjust the material thickness to fabricate it with different material. She also decides to make the model available on Thingiverse as it is versatile and robust to changes.

### 3.2 Extracting Relations

In order to suggest constraints, CODA continuously extracts the following four types of relations in CAD models [39]:

1. Ground relations: relations with respect to the reference coordinate system, such as a line being horizontal or vertical in a 2D sketch (Figure 3a).
2. Geometric relations: relations that define known geometric alignments, such as tangency, collinearity, parallelism, perpendicularity, and coincidence of points (Figure 3b).
3. Dimensional relations: relations between sizes or offsets between elements, such as edges with an equal length or a point in the middle of an edge (Figure 3c).

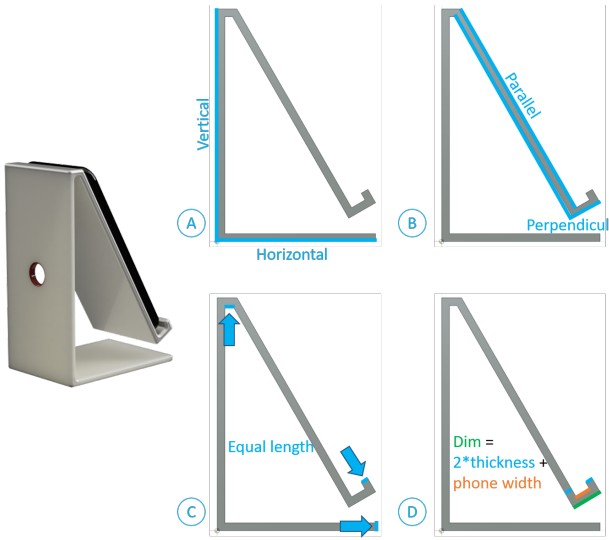

Figure 3: Cross section profile of a phone holder with four types of relations annotated: (a) Ground relations. (b) Geometric relations. (c) Dimensional relations. (d) Algebraic relations.

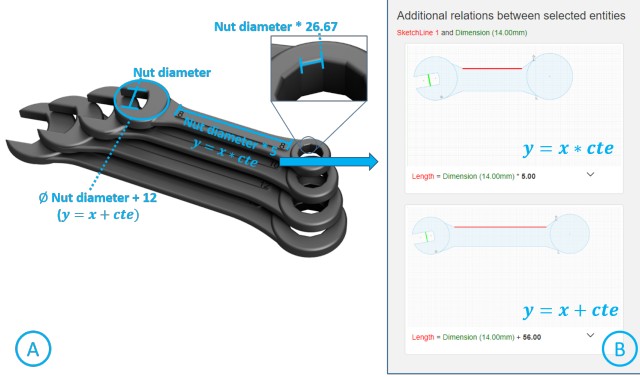

Figure 4: (a) This wrench requires custom constraints to ensure it scales appropriately with respect to the nut diameter. (b) CODA assist users in making such custom relations.

4. Algebraic relations: restrictions on the model in the form of mathematical expressions. For example, a edge being twice as long as another edge (Figure 3d).

While extracting relations, CODA considers parameters of modeling operations as well as attributes of all entities in a sketch (i.e. sketch entities). Sketch entities in Autodesk Fusion 360 include points, lines, circles, ellipses and arcs. Rectangles, for example, are not sketch entities but profiles as they consist of multiple lines. While CODA always extracts relations between exactly two attributes, attributes of various types can be related by CODA. Here we can distinguish the following combinations:

- Relations between sketch entities within a single sketch. Within a sketch, all four types of relations are applicable. For example, a line being tangent to a circle or a rectangle having twice the width of the diameter of a circle.
- Relations between sketch entities across different sketches. For these relations, dimensional and algebraic relations are applicable, such as a slot in two different sketches being equal in size.
- Relations between parameters of modeling operations. For these relations, dimensional and algebraic relations are applicable, such as two fillet operations with the same radius or the depth of an extrusion being equal or half the size of the radius of a fillet operation.
- Relations between a parameter of a modeling operation and a sketch entity. For these relations, dimensional and algebraic relations are applicable, such as the width of a slot in a sketch being equal to the depth of an extrusion.

To not overwhelm users with suggestions and to offer suggestions in context, we present relations within a single sketch only when the user is editing the respective sketch. All other relations are presented outside the 2D sketching mode in 3D modeling mode.

As the number of algebraic relations is possibly very large, especially when considering all pairs of sketch entities and parameters of modeling features, CODA first extracts algebraic relationships that are frequently present in CAD models. Studies of Mills et al. [32] and Langbein et al. [26] show that the most common relations in

CAD models include equal radii and lengths of edges, congruent faces, and radii and edges being half, one third, and one fourth in length. CODA thus presents algebraic relations between all pairs of sketch entities and parameters of modeling operations and vice versa that are equal, half, one third, and one fourth in value. To help users create well-constrained models with less common relations, the next section covers CODA's features to facilitate specifying custom algebraic relations between specific pairs of entities.

### 3.3 Custom Algebraic Constraints

To help users identify and specify custom algebraic constraints (Figure 4a), CODA supports an in-depth search between two entities. When starting this feature, the user selects two entities. These can be sketch entities (e.g. points, lines, circles, ellipses, arcs), dimensions, or modeling features (e.g. extrusion, fillet operations). CODA now calculates mathematical expressions between all pairs of attributes of the two entities, independent of whether the relation is a common ratio. As shown in Figure 4b, CODA suggests a relation for every pair of attributes both as ratios ($y = x * cte$) and sums ($y = x + cte$).

### 3.4 Explaining Suggested Constraints

To clearly communicate the meaning and implications of suggested constraints, CODA creates animated visualizations for all suggestions. Hovering a suggested constraint animates the model and communicates the meaning of the constraint in the context of the model. Explaining a constraint using animations on top of the model, in contrast to generic visualizations, textual explanations, or symbols, makes it convenient for users to understand its implications in the model. We developed two classes of animations to visualize the behavior of the four supported types of relations:

- For ground and geometric relations, we animate the variability currently present in the model when the suggested ground or geometric constraint would not be added. For example, hovering the suggested collinear constraint in Figure 2c, repeatedly moves the two edges that are collinear but not yet constrained as such. The animation makes the user aware that these two edges can still move with respect to each other. Using the "yes" and "no" buttons, the user specifies whether the demonstrated movement of these lines is allowed. "no" adds the collinear constraint while "yes" discards the suggestion without further action. Figure 5a gives an overview of how CODA animates all geometric and ground relations.
- For dimensional and algebraic relations, we animate how two entities would behave when their position or sizes would be

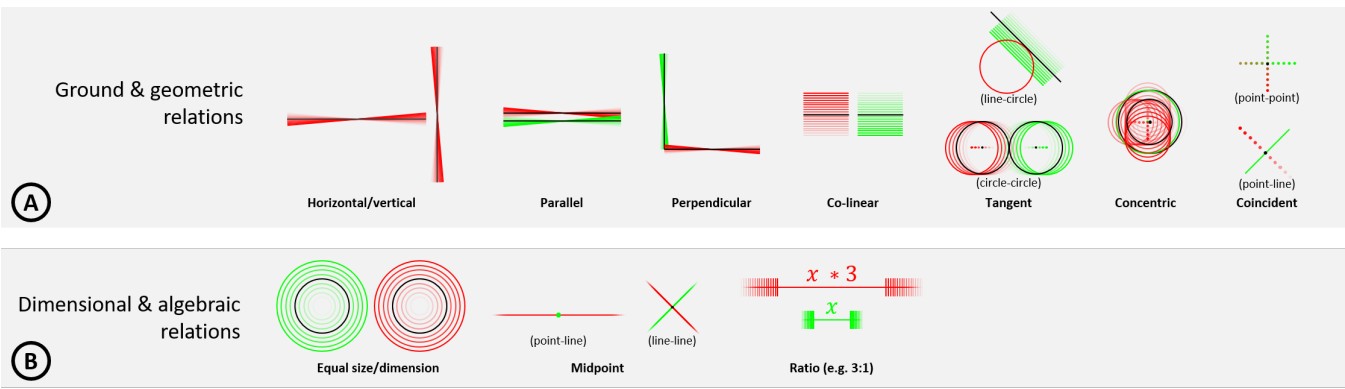

Figure 5: CODA animates the meaning and implications of suggested constraints. (a) shows an abstraction of animations for demonstrating ground and geometric relations. (b) shows an abstraction of animations for demonstrating dimensional and algebraic relations.

constrained to each other. For example, hovering the suggested dimensional constraint in Figure 2a and Figure 2b shows how two lines would scale when they are constrained to have equal dimensions. For these relations CODA asks whether the suggested relations fit the design. "yes" adds the respective dimensional or algebraic constraint, while "no" discards the suggestion. Figure 5b gives an overview of how CODA animates all dimensional and algebraic relations.

### 3.5 Invalidated Relations

It is important to note that CODA only suggests constraints to enforce relations that are present in the current version of the model. Oftentimes while testing the robustness of a model, the model breaks because of constraints that are still missing. Figure 6a shows how the symmetry in the laptop stand breaks when changing the width because of a missing constraint. These missing constraints will not be suggested by CODA as the relations are not present anymore in the broken model. To solve this inconvenience, CODA continuously presents a message communicating how many relations are invalidated by the last modeling operation (Figure 6b). As shown in Figure 6c, the user then gets the option to revert the last action that broke the model and is presented with the list of invalidated relations that could be enforced to further constrain the model. This is an iterative and powerful workflow that allows users to break relations in a model and see which constraints can be added to enforce these relations.

### 3.6 Navigating through Suggestions for Constraints

As CODA checks pair-wise for unconstrained relations between all entities within a sketch (in sketch-mode) and between entities across sketches and modeling features (in 3D modeling mode), the number of suggestions can grow rapidly. For example, six edges that are equal in length result in 30 suggestions to relate the lengths of all possible pairs of edges. Instead of presenting all these suggestions individually, CODA groups related suggestions into higher-level suggestions for constraints. As shown in Figure 7a, CODA suggests one constraint that links the lengths of all six edges. Accepting this suggestion adds all relevant individual constraints to the design. Such a higher-level suggestion can be expanded to see more details on what the suggestion entails (Figure 7b). In this detailed view, individual edges of equal length that CODA associated, can be excluded from the suggested constraint.

To further facilitate navigating through all suggested constraints, CODA presents suggestions per profile (Figure 1b). Selecting a profile shows all suggestions for constraints related to entities within this profile as well as constraints between entities in this profile

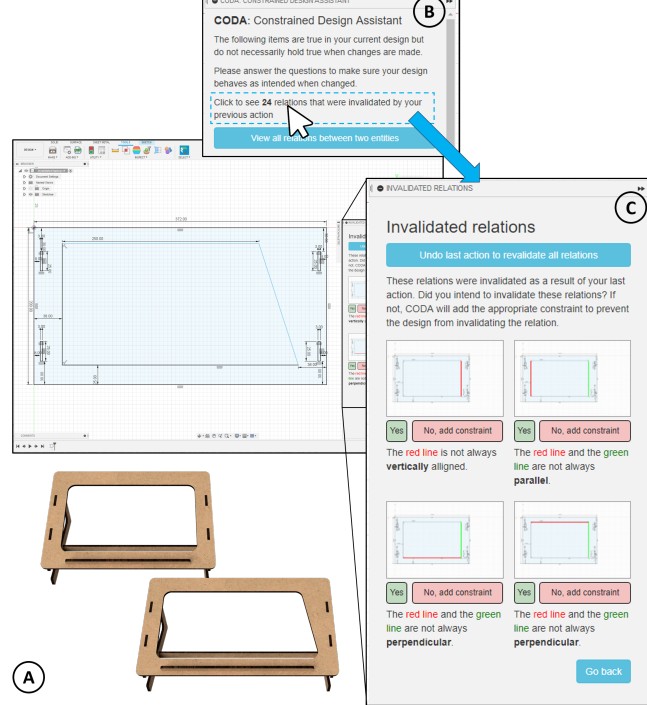

Figure 6: CODA only offers suggestions to enforce relations that are currently present in the model. (a) Changing parameters can break relations because of missing constraints. (b-c) CODA resolves this by showing a list of constraints that were invalidated by the last modeling operation.

and another profile. To filter the suggestions, users can also select multiple individual entities directly in the design after which CODA only shows the suggestions related to those entities.

## 4 IMPLEMENTATION

CODA is implemented as a Python plugin for Autodesk Fusion 360[8]. The concepts and features presented in CODA, however, are not specific to the Fusion 360 environment and could be imple-

[8]https://www.autodesk.com/products/fusion-360/overview

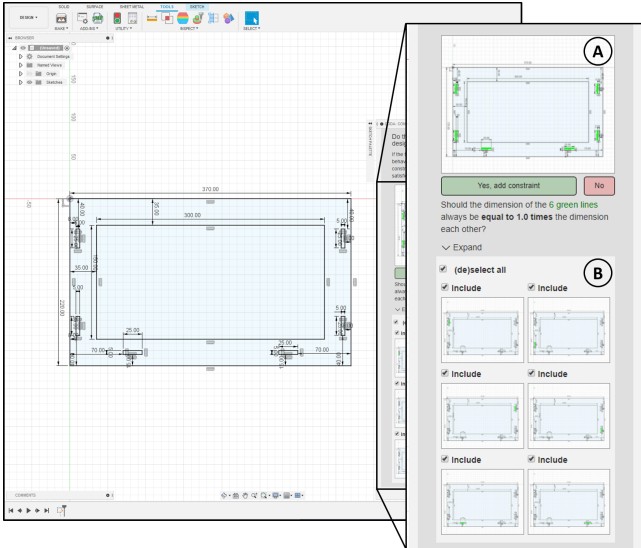

Figure 7: (a) CODA groups related suggestions into a higher-level suggestion. (b) By expanding, individual entities can be excluded from the grouped suggestion.

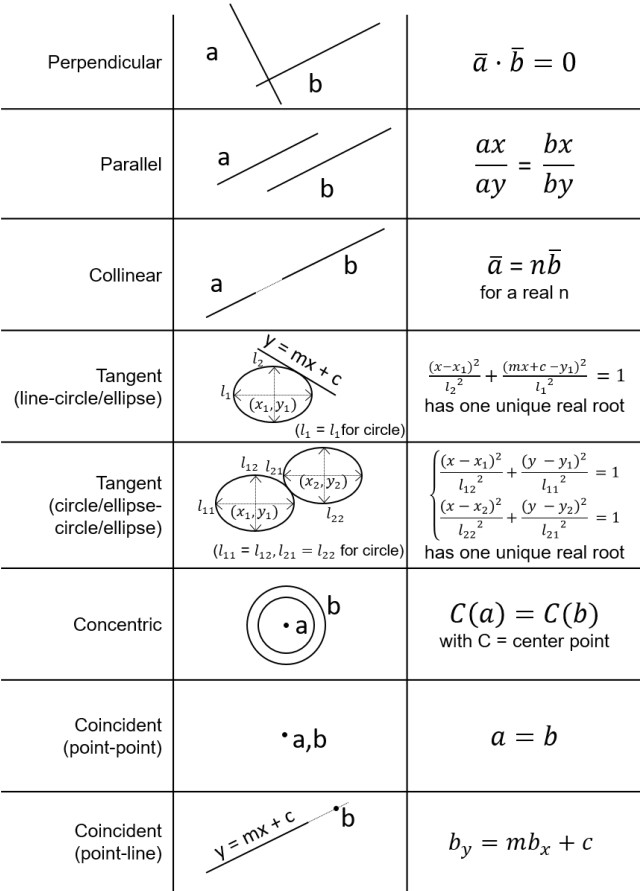

Figure 8: CODA uses basic linear algebra methods to check for geometric relations between entities in a sketch.

Table 1: Attributes of sketch entities considered by CODA to extract dimensional and algebraic relations.

| Sketch entity | Attributes |
|---|---|
| Points | (x,y) position |
| Lines | (x,y) position of start, mid, and end point, length of line |
| Circles | (x,y) position of center, diameter of circle |
| Ellipses | (x,y) position of center, size along minor and major axis |
| Arcs | (x,y) position of start, mid, and end point, radius of arc |

mented in other feature-based parametric CAD environments, such as SOLIDWORKS, Rhinoceros 3D, or Autodesk Inventor.

## 4.1 Extracting unconstrained relations

To suggest constraints, CODA continuously checks for the presence of unconstrained relations between all pairs of sketch entities within a single sketch and across different sketches. These calculations are done in a separate thread to ensure the interface always remains interactive. As long as the user is making edits, CODA re-initiates these calculations once the previous thread is finished. Figure 8 gives an overview of how CODA checks if a geometric relation is present between two sketch entities.

When extracting dimensional and algebraic relations, CODA finds pairs of attributes in a 3D CAD model that are equal, half, one third, and one fourth in value [26, 32]. Note that for these relations, parameter values of modeling operations, such as extrusions and fillets, as well as the position and sizes of sketch entities are considered. Table 1 gives an overview of the attributes CODA considers for extracting dimensional and algebraic relations per sketch entity.

For CODA to only suggest relations that are not yet enforced by constraints, our algorithm needs to take into account constraints already present in CAD models. The Fusion 360 API, however, only exposes constraints that are explicitly present in the CAD model. Sketch entities, however, can also be implicitly constrained by other constraints. The next subsection discusses how we extract those implicit constraints to ensure CODA does not offer these suggestions.

## 4.2 Extracting implicit constraints present in the model

Fusion 360's API does not provide access to the constraint graph and only exposes constraints that are explicitly added by the user or through the API. Besides these constraints, however, other implicit constraints can be present in a sketch. For example, two lines with a parallel constraint to a third line are always parallel to each other without that parallel constraint being present. CODA needs to be aware of these implicit constraints to avoid suggesting constraints that are already present in the model (Section 4.1) as well as to prevent over-constraining models (Section 4.3).

To extract these implicit constraints, we re-implemented and extended the technique of Juan-Arinyo and Soto [22]. This technique requires all constraints to be expressed as either constrained distance (CD) sets, constrained angle (CA) sets, or constrained perpendicular distance (CH) sets. While CD sets includes points between which all distances are constrained, CA sets consist of line segments between which all angles are constrained, and CH sets consist of a point for which the perpendicular distance to a line segment is constrained. We convert all length/size constraints in Fusion 360 to constrained distance (CD) sets, angle constraints to constrained angles (CA) sets, and offset constraints to constrained perpendicular distance (CH) sets. For ground and geometry constraints in Fusion 360 we use the following conversion:

- Horizontal/vertical: We add a constrained angle (CA) set rep-

resenting an angle of 0° between the line segment and the x- or y-axis.

- Perpendicular: We add a constrained angle (CA) set representing an angle of 90° between the two line segments.
- Parallel: We add a constrained angle (CA) set representing an angle of 0° between the two line segments.
- Collinear: For all pairs of end-points of the two line segments, we add a constrained perpendicular distance (CH) set with a distance of 0.
- Concentric: We add a constrained distance (CD) set, representing a distance of 0 between the mid-points of the two concentric circles.
- Coincident (point-point): We add a constrained distance (CD) set, representing a distance of 0 between the coincident points.
- Coincident (point-line): We add a constrained perpendicular distance (CH) set, representing a distance of 0 between the point and the line.
- Tangent (circle/ellipse-line): We add a constrained angle (CA) set, representing an angle of 90° between the line segment and the radius of the circle at the tangency point.
- Tangent (circle/ellipse-circle/ellipse): We add a constrained angle (CA) set, representing an angle of 0° between the radii of the two circles at the tangency point.

Once all Fusion constraints are converted to CD, CA, and CH sets, we compute the transitive closure of the constrained angles (CA) sets and use the 20 rules presented by Juan-Arinyo and Soto [22] to merge constraints. Now we get CD, CA, and CH sets that reflect the implicit constraints. When two sketch entities have a relation that is not yet enforced by a explicit constraint, CODA can now check whether these entities are implicitly constraint using the following rules:

- Horizontal/vertical: Implicitly constrained if a constrained angle (CA) set exists between the line segment and a line segment representing the x- or y-axis.
- Perpendicular/parallel: Implicitly constrained if a constrained angle (CA) set exists representing both line segments or if a constrained distance (CD) set exists representing the four points of the two line segments.
- Collinear: Implicitly constrained if a constrained perpendicular distance (CH) set exists representing both line segments or if a constrained distance (CD) set exists representing the four points of the two line segments.
- Concentric: Implicitly constrained if a constrained distance (CD) set exists representing the center points of the two circles.
- Coincident point-point: Implicitly constrained if a constrained distance (CD) set exists representing the two points.
- Coincident point-line: Implicitly constrained if a constrained perpendicular distance (CH) set exists representing the point and the line or if a constrained distance (CD) set exists representing the point and both points of the line segment.
- Tangent circle/ellipse-line: Implicitly constrained if a constrained angle (CA) set exists representing the line segment and the radius (line segment) passing through the tangency point. Alternatively, if a constrained distance (CD) set exists including the two points of the line segments and the two points of the radius passing through the tangency point.
- Tangent circle/ellipse-circle/ellipse: Implicitly constrained if a constrained angle (CA) set exists representing the radii of both circles passing through the tangency point. Alternatively, if a constrained distance (CD) set exists including the four points of the two line segments passing through the tangency point.

### 4.3 Removing dimensions to avoid over-constraining models

Accepting suggested constraints oftentimes requires removing existing constraints present in the model. In the sketch in Figure 9a-left,

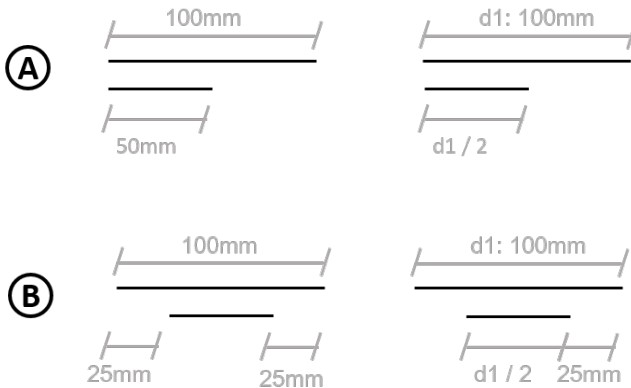

Figure 9: When accepting suggested constraints CODA oftentimes (a) overwrites existing constraints or (b) removes existing constraints.

for example, CODA suggests to constrain the length of the lines so that they are always half the length of each other. To accept this constraint, CODA replaces the static dimensional constraint with the dynamic constraint shown in Figure 9a-right. However, when both dimensions are implicitly constrained as shown in Figure 9b-left, CODA needs to remove one of the other constraints before adding the suggested constraint shown in Figure 9b-right. For CODA to know the explicit constraints that are responsible for every implicit constraint, we keep track of all explicit constraints while merging CD, CA, and CH sets in the algorithm explained in Section 4.2. When multiple explicit constraints are responsible for an implicit constraint (Figure 9b), CODA shows multiple suggestions and communicates their differences through the animated visualizations.

### 4.4 Rendering animations in CAD models

When hovering suggested constraints, CODA previews the implications of a constraint by animating features in the CAD model (Figure 5). When the animation requires lines to tilt, we continuously rotate the line between -5° and +5°. When changes in size are required, we apply a scaling that transitions between half and double the size of the entity. Finally, when animating a parameter of a modeling operation, such as an extrusion, we alternate between half and double of the original value. Animations are updated every 100ms and are repeated as long as the user hovers the suggested constraint.

As CODA directly manipulates parameters in the original design, we make a copy of all the attributes to be able to restore them afterwards. For some sketch entities, additional temporary construction lines are required to realize the animation. For example, to vary the angle between two parallel lines, Fusion 360 does not allow to temporarily add an angular constraint between the two lines as they are exactly parallel. CODA therefore first adds a temporary construction line perpendicular to both lines and varies the angle between the perpendicular line and both parallel lines during the animation (Figure 5a).

### 5 BENCHMARKING THE NUMBER OF SUGGESTIONS

The number of suggestions offered by CODA depends on the number of relations present in the model that are not yet constrained. Furthermore, while designing, this number changes continuously as new relations are established or broken and constraints are added. To give the reader insights in how many suggestions CODA offers, we report the number of suggestions for the CNC milling plans of a

Table 2: The number of unique high-level suggestions provided by CODA on three real-world CNC-millable furniture models from opendesk.

| Model | Sketch | Condition | # sketch entities | # constraints | Suggestions |
|---|---|---|---|---|---|
| Lean desk | Tabletop | No constraints | 52 | 0 | 46 |
| | | Default constraints | 52 | 37 | 14 |
| | Support beam 1 | No constraints | 74 | 0 | 52 |
| | | Default constraints | 74 | 25 | 26 |
| | Support beam 2 | No constraints | 38 | 0 | 32 |
| | | Default constraints | 38 | 16 | 20 |
| | Main beam | No constraints | 208 | 0 | 239 |
| | | Default constraints | 208 | 65 | 204 |
| | Outer leg | No constraints | 169 | 0 | 71 |
| | | Default constraints | 169 | 46 | 68 |
| | Inner leg | No constraints | 159 | 0 | 87 |
| | | Default constraints | 159 | 30 | 73 |
| | Cable rail support | No constraints | 44 | 0 | 22 |
| | | Default constraints | 44 | 15 | 11 |
| Fin bookshelf | Shelf | No constraints | 106 | 0 | 109 |
| | | Default constraints | 106 | 31 | 88 |
| | Back | No constraints | 581 | 0 | 490 |
| | | Default constraints | 581 | 173 | 401 |
| | Divider | No constraints | 106 | 0 | 120 |
| | | Default constraints | 106 | 33 | 84 |
| | Side | No constraints | 152 | 0 | 139 |
| | | Default constraints | 152 | 42 | 116 |
| | Top | No constraints | 161 | 0 | 140 |
| | | Default constraints | 161 | 46 | 89 |
| | Bottom | No constraints | 315 | 0 | 241 |
| | | Default constraints | 315 | 78 | 160 |
| | Foot part 1 | No constraints | 44 | 0 | 59 |
| | | Default constraints | 44 | 12 | 56 |
| | Foot part 2 | No constraints | 44 | 0 | 61 |
| | | Default constraints | 44 | 13 | 46 |
| Slim chair | Leg | No constraints | 48 | 0 | 24 |
| | | Default constraints | 48 | 17 | 12 |
| | Leg join | No constraints | 32 | 0 | 12 |
| | | Default constraints | 32 | 12 | 9 |
| | Backrest | No constraints | 42 | 0 | 45 |
| | | Default constraints | 42 | 16 | 29 |
| | Backrest join | No constraints | 54 | 0 | 30 |
| | | Default constraints | 54 | 22 | 22 |
| | Seat | No constraints | 34 | 0 | 29 |
| | | Default constraints | 34 | 14 | 17 |

lean desk, a fin bookshelf, and a slim chair available on Opendesk[9] (Table 2). We ran CODA on all sketches present in the three models. As the plans are available in the DXF file format, the original sketches do not have any constraints. The "No constraints" condition in Table 2 thus reports on the number of suggestions one gets while transferring a completely unconstrained sketch to a parametric sketch. However, when using recent versions of CAD modeling environments, such as Fusion 360, a sketch without any constraints is very uncommon as CAD environments automatically add basic constraints while sketching. To get an idea of the number of suggestions in a realistic modeling scenario, we also re-designed all sketches in Fusion 360. The "Default constraints" condition in Table 2 reports on the number of suggestions offered by CODA once these sketches were finished. The number of suggestions reported in this table use the grouping strategy presented in Section 3.6. It is also important to mention that the numbers reported here are for completed sketches, some of which are complex. We hope, however, that users will not use CODA only at the end of a design workflow but also consider,

[9]https://www.opendesk.cc/

accept, and discard suggestions during the modeling workflow. As such, suggestions are gradually introduced and will not accumulate. When users are overwhelmed with suggestions, filtering and selection features can be used as discussed in Section 3.6.

## 6 LIMITATIONS AND FUTURE WORK

Although CODA offers many novel opportunities to facilitate making well-constrained CAD models, our work also has several limitations which reveal many exciting directions for future research.

First, future research could study how novices in CAD modeling use CODA during their design workflow. While CODA offers suggestions in real-time while modeling, our tool could also be used after a CAD model is finished or at a later time to make an existing model more flexible and adaptable. We believe this could be a major asset as it allows novices to further improve CAD models shared via platforms, such as Thingiverse, and thus distribute the workload across the community. Furthermore, it would be interesting to investigate whether CODA also provides value to expert modelers and for students that learn CAD modeling. For example, the suggested constraints can make students aware of available CAD features and

how they are composed.

Second, while CODA offers suggestions for the most common constraints in CAD, more advanced suggestions can be supported in the future. For geometry relations this includes identifying patterns, such as symmetry or repetition in models and offering suggestions to convert these patterns into more adaptable features. For algebraic constraints, the current version of CODA supports frequently used ratios in CAD models according to Mills et al. [32] and Langbein et al. [26]. Other common ratios used in design could be supported in the future, such as the Golden ratio or the Lichtenberg ratio. Future versions could also offer suggestions for relations that are nearly present in the model, such as lines that are almost perpendicular or are almost equal in length. Similar approaches have been explored for beautifying mesh models [24].

Third, to further facilitate adapting models, future versions of CODA could offer suggestions for other types of constraints, such as limiting the range in which parameters and dimensions can change without breaking the model. While computing valid ranges of features has been investigated for 2D sketches [13, 14, 41], more research is needed to compute valid ranges for all features in 3D to ensure the integrity of the model when changing parameters.

## 7 CONCLUSION

In this paper we presented CODA, an interactive software tool that helps novice modelers to design well-constrained parametric 2D and 3D CAD models. In order to do so, CODA contributes a computational approach for extracting and suggesting relations in a model that are not yet enforced by constraints. CODA also clearly communicates the meaning and implications of suggested constraints using novel animated visualizations rendered in the CAD model. By facilitating the creation of well-constrained parametric designs we hope to further democratize CAD and encourage users to upload high quality parametric models to public sharing repositories, such as Thingiverse.

### ACKNOWLEDGMENTS

We thank Danny Leen and Kris Luyten for their valuable input in this work, and Thijs Roumen and Tovi Grossman for the fruitful discussions. This research was supported by the Special Research Fund (BOF) of Hasselt University.

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
