# OpenReview forum: "CODA: A Design Assistant to Facilitate Specifying Constraints and Parametric Behavior in CAD Models"
_graphicsinterface.org/Graphics_Interface/2021/Conference — GI 2021_

### Official Review · AnonReviewer2 · 2021-01-11
**The paper presents an interface to help aid novice users in parameterizing CAD models. Simple examples throughout the paper help demonstrate its utility in designing well-parameterized CAD models. On the negative side there is no formal evaluation of the proposed tool, making it unclear how well it works in practice or how it scales to more complex CAD models.**

**Rating:** 6
**Confidence:** 3

**Review:**

While parameterized CAD models allow for quick adaptation and redesign of an existing model, authoring well-parameterized models is challenging – especially for novice users (e.g. hobbyists from the maker community). The paper presents an interactive software tool to aid in the design of CAD models, which aims to help novice users create well-parameterized models, by identifying unconstrained relations, and presenting them to the user via interactive animations.

Overall, how the system is implemented and would be used is well-explained. The design of the system (e.g. how constraints are identified and presented to the user) appears grounded in the literature on how CAD software is used and the difficulties users encounter in creating parametric models. The presented case studies, including the design of the laptop stand, provide a tangle impression of how the system could be used to streamline the process of constraining parametric models.

That being said, I still find it unclear how well the system works in practice. It appears that the main novel contribution of the work lies in interface design. Without some form of formal evaluation of the tool’s usability it is difficult to tell how well CODA achieves its intended goal – aiding novice users in the design of well parameterized CAD models. This problem is made more pronounced by the fact that many of the references used to support design decisions seem to be based off the work of experienced CAD users (e.g. Millis et al., Langbein et al).

I am also concerned by the scalability of the interface to more complex models, it’s unclear for me that users would be able to effectively navigate the increasing number of automatically generated constraints to create well-parameterized models. Some statistics on the number of constraints identified and presented to users for a selection of models may be helpful in this regard. Similarly, I’d like to see some runtime information (e.g. how long it takes to generate constraints, and how interactivity scales with model complexity).

Also, the substantial literature on extracting constraints in the context of reverse engineering makes it harder to appreciate the approach for extracting unconstrained relations as a contribution. Some comparison of the proposed approach to pre-existing works in this context would strengthen the paper.

---

### Official Review · AnonReviewer1 · 2021-01-13
**The paper presents an interactive software tool for facilitating the design process of CAD models by proposing constraints to novice users. While the tool seems practical and well implemented, there is no technical novelty in finding and proposing new relations. Furthermore, it is unclear how well it scales to more complex designs.**

**Rating:** 6
**Confidence:** 3

**Review:**

The paper presents an interactive software tool for facilitating the design process of CAD models by proposing constraints to novice users. This user interface is available as a plugin for Autodesk Fusion 360.

The paper is well-written and easy to follow. It also presents an extensive overview of the related work. The problem that is being addressed is interesting and useful. Implementation and workflow are well explained.

The main contribution of the paper seems to be an intuitive user interface for CAD modeling. While the paper gives solid examples of the use case, a user study would improve the quality of the work. Furthermore, there is not much novelty in how to search for and how to compute relations for constraints.

Another downside is that the presented examples look too simplistic. It would be good to show a more complex sketch. It is unclear how fast the CODA computes and proposes constraints. How does the computation time increase with the object complexity?

---

### Official Review · AnonReviewer3 · 2021-01-13
**Nice paper; no validation**

**Rating:** 6
**Confidence:** 4

**Review:**

The paper presents a new assistant tool for Autodesk Fusion 360, target at novice users, that notices regularities (equal sizes, parallelism, ratios, etc) and suggests to add those as constraints. These regularities can be processed either within a single sketch or across a collection of sketches. Once those constraints are in, the model can be parameterized and then reused in, say, Customizer. This simplifies the creation of parametric models and hopefully will increase their percentage in datasets like Thingiverse.

I personally loved the paper: it's clearly written, solves a very well defined task, addresses an important problem (it is indeed very hard to create parametric models), and has quite a few clever ideas along the way. For example, the idea to use previous statistical analysis to understand the more used types of relations/regularities people use and suggest only those can be very powerful, even if the idea itself is straightforward. My only complain on the text is that it's a little repetitive and can be shorter (for example, beginning of the related work repeats word by word phrases from the intro, and 4.1 repeats parts of 3.2).

My only real concern, however, is complete absence of validation. I would have expected one or two user studies proving that the system is a) usable and b) saves time designing parametric models. This is the only reason why I'm not giving a higher score, other than that I'm quite sure this is useful and well done.

---

### Meta-Review · Area_Chair1 · 2021-01-14

**Recommendation:** Accept
**Confidence:** 4

**Metareview:**

The recommendation for this paper is acceptance. All reviewers agree that the paper tackles an important problem and is well-written. Several concerns that were raised and that should be addressed in the revised version include: there is no validation in the paper (a user study should be conducted), a more complex example should be presented, some runtimes should be included, and the paper could be shortened by removing repetitive parts (some parts a repeated word-by-word).

---

### Decision · Program_Chairs · 2021-01-16

Accept